# Food Allergens: When Friends Become Foes—Caveats and Opportunities for Oral Immunotherapy Based on Deactivation Methods

**DOI:** 10.3390/nu15163650

**Published:** 2023-08-20

**Authors:** M. Victoria Gil, Nuria Fernández-Rivera, Carlos Pastor-Vargas, Pedro Cintas

**Affiliations:** 1Departamento de Química Orgánica e Inorgánica, Facultad de Ciencias, IACYS-Unidad de Química Verde y Desarrollo Sostenible, Universidad de Extremadura, 06006 Badajoz, Spain; 2Departamento de Bioquímica y Biología Molecular, Universidad Complutense de Madrid, 28040 Madrid, Spain

**Keywords:** food allergy, cow’s milk proteins, phenolic compounds, circular chemistry

## Abstract

Food allergies represent a serious health concern and, since the 1990s, they have risen gradually in high-income countries. Unfortunately, the problem is complex because genetic, epigenetic, and environmental factors may be collectively involved. Prevention and diagnoses have not yet evolved into efficacious therapies. Identification and control of allergens present in edible substances hold promise for multi-purpose biomedical approaches, including oral immunotherapy. This review highlights recent studies and methods to modify the otherwise innocuous native proteins in most subjects, and how oral treatments targeting immune responses could help cancel out the potential risks in hypersensitive individuals, especially children. We have focused on some physical methods that can easily be conducted, along with chemo-enzymatic modifications of allergens by means of peptides and phytochemicals in particular. The latter, accessible from naturally-occurring substances, provide an added value to hypoallergenic matrices employing vegetal wastes, a point where food chemistry meets sustainable goals as well.

## 1. Introduction

In a broad sense, the terms allergy and hypersensitivity are usually employed to describe inappropriate immune responses after exposure to substances that are harmless in most subjects. Although the immune system can respond to almost any foreign molecule, there is little doubt that food allergies represent a dominant manifestation of hypersensitivity reactions and a major health concern worldwide affecting both adults and children [1,2,3,4,5]. It is well established that this type of atopic or anaphylactic hypersensitivity occurs when the antigen (or specific allergens present in food) reacts with IgE antibodies bound to tissue mast cells and triggers the extrusion of granules from such cells. This mechanism causes the release of granular contents into the surrounding fluids, namely histamine, heparin, or enzymes causing potent local effects on muscles and blood vessels, among others. Moreover, the stimulated mast cells make other chemicals like prostaglandins and leukotrienes, which enhance the above effects further. Overall, the antigen-IgE-mast cell reaction is responsible for acute inflammation. While this process can be a local protective mechanism, the manifestations of an allergic reaction may be of variable intensity, ultimately ending up in life-threatening anaphylactic shock.

Although the stepwise reaction of the immune system is well characterized, the ultimate origin of food allergies and the molecular processes brought into play have not been fully explained. A discussion of such mechanisms lies beyond the scope of this review, yet a few recent and emerging hypotheses and facts should be highlighted in context [6]. Thus, we know that allergens can be transferred in the womb [7], and fetal mast cells may contribute to antigen-specific vertical transmission through the developing fetus [8,9]. On the other hand, the gut microbiome needs to be reassessed in terms of mechanisms triggering immune responses, but also as protective routes against food allergies [10,11]. Gut microbial composition of ages three to six months has pronounced effects on food allergy outcomes [12]. Identification of a set of microbial species that promote the development of a stable immune-protected gut environment could make it possible to envisage protective microbiome-oriented approaches for reversing the onset of allergy [13]. The impact and variability of food allergies are diverse, although they are mostly caused by cow milk, egg, and peanuts in children (Figure 1) [14].

For patients and children in particular, the pathological scenario of physical symptoms is often worsened by psychological constraints affecting mental health, as they are forced to follow strict routine protocols that involve, among others, preparing allergen-free meals with expensive ingredients, reading food labels, and controlling social encounters in schools and restaurants. Bullying, anxiety and depression can have a huge impact, and recent reports indicate that one in three children with food allergies is bullied as a result of this disease [15,16]. In context, the COVID-19 pandemic has also impacted the way people eat and demand takeaway food [17]. Not by chance, food allergy has been rightly denoted as “everyone’s business” [18], and the United States President, Joe Biden, signed recently The Food Allergy, Safety, Treatment, Education, and Research Act [19], aimed at improving data collection on the prevalence and severity of food allergies, while promoting research and development for diagnosis and prevention.

For decades, the long-held and otherwise effective therapeutic treatment has involved habitual abstaining from foods containing the proteins responsible for allergy. In recent years, the ever-growing approach of oral immunotherapy treatments (OIT) has gained considerable attention, yet it shows both pros and cons. In short, OIT consists of the oral administration of increasing doses of the allergen until reaching the “target dose”. This stage of the treatment is called the “up-dosing phase”, which lasts until the so-called “maintenance phase”, i.e., maintaining the target dose as the regular dose [20,21]. The way the oral treatment works in molecular terms is still a complex issue, not yet elucidated in detail, and presumably, other Ig species and the intermediacy of T cells are involved. Thus, immunotherapy could lead to the production of the IgG4 antibody, which is thought to compete with IgE and prevent mast cells from being activated. Concomitant production of another antibody (IgA) would make it harder for the allergens to pass from the gut into the bloodstream (Figure 2). As a result, less food protein is available in the bloodstream and fewer mast cells are generated, thereby limiting the release of endogenous markers such as histamine [14].

Cow’s milk protein allergy (CMPA) is one of the most frequent pathologies in early life having a prevalence of about 0.5–3% in the first year of life [22]. This allergy can also come from the breast milk of mothers who are consuming cow products as well as infant formulas [23]. For the sake of clarity, the World Allergy Organization (WAO) established that cow’s milk allergy (or allergic milk hypersensitivity) refers specifically to an IgE-mediated response accompanied by symptoms such as atopic eczema, rhinitis and/or asthma. In addition, the reaction could be non-IgE mediated and would include symptoms linked to gastrointestinal disorders [24]. Cow’s milk is composed of about 30 proteins, all being potential allergens. Some are relatively insoluble, such as casein, and are found in the fraction called coagulum, while others are present in lactoserum (β-lactoglobulin or β-lactalbumin). For such a broad range of proteins, different epitopes have been found and are likely responsible for allergenic reactions [25]. There have been some systematic analyses and meta-analyses of food allergies caused by common foods in developed countries [4], which provide a broad range of prevalence rates depending on the type of food. Incidences are strongly influenced by the diet through different countries and zones, as portrayed by the EuroPrevall serum bank [26,27].

It is worth pointing out that OIT can be circumscribed within the range of immunotherapy treatments (IT) [28], all involving the gradual administration of allergens and differing in the type of allergen delivery (Figure 3). Certainly, this classification may be somewhat misleading (oral versus sublingual, for instance), albeit the applicability varies from patient to patient and early detection of side effects represents the bottleneck to adopt OIT as a general trend [29].

Thus, OIT against milk allergy has gained attention in recent years using different doses of cow’s milk directly. Clinical observations, however, detected risks in some individuals susceptible to allergic reactions or unable to complete prolonged treatments [30]. Similar OIT studies have been conducted against egg- and peanut-caused allergies [31,32], which disclose related unwanted effects. Clearly, peanut allergy affects largely children in the United States, and the Food and Drug Administration approved the first drug to treat such an allergy, an OIT called Palforzia [33]. Improving the safety of OIT represents a major challenge, yet some strategies are well identified with further pursuits underway. These include the aforementioned alternative delivery of allergens, the use of adjuvants (probiotics, IgE inhibitors, etc.) that mitigate the immune response, or modified hypoallergenics [34,35]. The latter constitutes the main focus of this review article, namely the approaches employed to produce either less harmful food proteins or hypoallergenic matrices combining proteins and other biomolecules.

At first sight, fighting proteins that cause a pathological situation could be reminiscent of targeting proteins involved in diseases for elimination. In context, the PROTAC (Proteolysis Targeting Chimera) concept harnesses the cell’s natural protein-degradation machinery (ubiquitin-proteasome system) to that end. Currently, nearly 20 protein degraders have entered clinical trials against various tumor types [36]. Unlike conditions that cause tumor growth, however, allergens are friendly agents in most subjects and will only induce abnormal responses in hypersensitive people. Deactivation rather than degradation should accordingly be the way of attenuating the immune response. Methods involve both physical and chemo-enzymatic routes. The potentiality, advantages and limitations will be illustrated through representative cases, especially those documented in recent times.

## 2. Physical Deactivation

### 2.1. Thermal Treatment

Heating is surely the oldest and most common protocol to modify the structure and function of native proteins [37]. It is pertinent to note that even baked matrices may increase tolerance to food allergens by modifying their structure, as evidenced in the case of muffins containing egg and peanut allergens [38]. Food cooking and processing will inevitably meet the ubiquitous Maillard reaction, the non-enzymatic browning involving the condensation of proteins and carbohydrates. Although multiple reaction products have been identified in Maillard reactions, which are influenced by numerous factors like temperature, pH, and substrates [39], in general, the glycation of individual proteins alters their functionality and in vivo behavior. As a result, digestibility, bioavailability, and immune response, including allergenicity, are affected as well. Thermal treatment, however, does not always lead to a hypoallergenic material. Thus, several studies on β-lactoglobulin, a key cow’s milk protein, indicate that heating and glycation caused by Maillard reactions with mono- and disaccharides results in a greater inertness toward proteolysis, thereby increasing the allergenicity [40].

Maillard reactions may affect the way specific IgE binds to food allergens. This can be achieved through different mechanisms such as (a) conformational disruption of secondary and tertiary structures that impair the IgE binding potential of the protein; (b) formation of agglomerates that enhance degranulation and/or enzyme liberation from basophils, and (c) formation of new epitopes (i.e., part of the antigen recognized by the immune system) due to aggregation and side Maillard reactions (Figure 4). 

Dry heating of cow’s milk protein in the presence of lactose leads to losses in solubility and digestibility, casein being the most altered protein. At low temperatures (60 °C), the decreased solubility is mainly induced by H-bonding and hydrophobic interactions, without impairing protein hydrolysis nevertheless. At high temperatures (130 °C), covalent protein cross-linking arises from Maillard reactions to a great extent, and is, however, responsible for poor digestibility and low solubility [41].

The aggregation trend of thermally treated codfish parvalbumin evidences an extensive ligation of lysine residues with glucose. However, this protein surface modification does not affect the folding nor significantly impair calcium binding. Glucosylation likely results in a lower hydrophobicity of the denatured state that slows down the aggregation process. In fact, aggregation and shape are comparable to those of the unmodified protein, and the resulting material undergoes faster digestion, thereby pointing to an effective treatment against allergic responses [42]. In a related study, the allergenicity of tropomyosin, the prevalent shellfish allergen, is reduced (up to 60%) through Maillard reactions with oligosaccharides. Results are consistent with conformational changes of the protein epitopes, with glycation leading to α-helix disruption as the main mechanism accounting for allergenicity reduction [43].

Heating along with other physical treatments (i.e., high pressure, ultrasound or electric fields, which we shall see later), can unfold native egg white proteins that improve their hydrolysis and digestibility. However, Maillard reactions have apparently opposite effects by reducing the digestibility and also lowering the IgE binding of ovalbumin, the latter potentially alleviating allergenicity. Supramolecular analyses suggest protein-protein interactions between unfolded states leading to aggregates of varied morphologies depending on the experimental conditions. Heat-induced spherical aggregates show lower accessibility of digestive enzymes than linear counterparts. However, higher supramolecular networks (i.e., gels) from linear aggregates are likewise reluctant to undergo enzymatic digestion [44].

Conformational changes are also behind the thermal deactivation of ovotransferrin, with reversible unfolding occurring at 55–60 °C. As temperature increases, the secondary structure is progressively disrupted and covalent disulfide bonds are cleaved above 80 °C. Overall, heating appears to have a positive effect by eliminating potentially allergenic epitopes of the protein [45]. Analyses of cow’s milk and hen’s egg white proteins reveal different patterns of allergenicity after heating. Most proteins become weaker, although ovomucoid remains stable. Time and temperature are key variables and the presence of wheat during heating decreases the IgE binding to proteins [46]. 

Despite numerous and often inconclusive statements, the cautionary lesson inferred from a recent review is that food (thermal) processing can either hide epitopes or change conformations of native proteins, though it is insufficient to safely protect from allergenic responses [47]. In that sense, another product used to improve the efficacy and safety profile in oral food challenges is dehydrated egg white. The allergenicity of commercially available dehydrated egg white is equivalent to raw egg white, but the former avoids microbiologically risk [48].

### 2.2. High-Pressure Treatment

High-pressure processing (HPP) emerges as a non-thermal procedure, sometimes an alternative to thermal treatment. Its applicability as a sterilizing method against microorganisms and foodborne pathogens is well established. Foods are exposed to high pressures, usually in the range of 300–700 MPa (from nearly 3000 to 7000 atm) for short periods (from seconds to a few min), with structural variations in biomolecules involving disruption of H-bonds and saline/dipolar interactions, hydrophobic interactions, and even weak covalent bonds, all resulting in conformational changes and protein denaturation [49,50,51]. The effects of HPP upon the Maillard reaction are complex indeed, and it would be incorrect to state that high pressure alone reduces the unwanted changes associated with browning. However, effects on protein unfolding have been documented [39,52]. The combined use of HP and high-temperature processing has been reported to retard Maillard reactions in whey protein-glucose model solutions [53].

HP processing of milk alleviates atopic dermatitis in a mouse model. Animals show lower levels of IgE in serum compared with untreated mice. Moreover, HPP decreases the cytokine production of T cells, especially Th1 and Th2 types, although no significant variations are observed for IL-2, IL-6, and IL-17A cytokines [54]. The effect of HP (at 550 MPa) on β-lactoglobulin evidences notable conformational changes, although the digestibility is not seriously compromised. The lower antigenic response seems to be caused by the hiding of conformational epitopes, which result from aggregation under hydrostatic pressure [55]. A comparative analysis between thermal pasteurization and HPP on bovine milk indicates that β-lactoglobulin and IgG undergo denaturation within the range of 400–600 MPa, while others (casein micelles) show minor conformational changes. The immunogenicity increases up to 600 MPa (at 30 °C for 15 min) due to protein aggregation, while at 600 MPa the secretion of Th-type cytokines diminishes, which can be related to hidden epitopes caused by sequential protein unfolding and aggregation. In contrast, thermal pasteurization (72 °C for 15 s) has little or no effect on immunogenicity [56]. The allergenicity of α-casein is reduced by means of HP (200–600 MPa), although a more pronounced effect can be induced by ultraviolet-C radiation. Changes in α-helicity, β-turn, and protein hydrophobicity appear to be the main structural motifs affected [57]. The allergenicity of HP-treated ovomucoid, based on the liberation of β-hexosaminidase from human prebasophils sensitized with sera from allergic patients, is reduced as pressure increases from 0.1 to 400 MPa. The higher figures are 88.4% and 80.7% inhibition at 400 and 500 MPa, respectively. Irreversible structural changes are associated with the unfolding of the tertiary structure of ovomucoid by exposing hydrophobic sites at the surface, whereas polar domains (tyrosine side chains in particular) increase [58]. HPP of ovoalbumin, compared with thermal treatment, induces structural deformations that are more significant in dilute aqueous solutions than in concentrated protein samples. Under hydrostatic pressure, water molecules will be able to disrupt the secondary structure by altering the hydrophobic interactions, which are however maintained in condensed phases [59]. It is worth pointing out that egg lysozyme exhibits enhanced catalytic activity under pressurization, albeit the pressures employed are much lower (ca. 150 atm) than those commonly used in food processing. This effect can however be advantageously harnessed in biocatalytic applications of enzymes [60].

HPP of almond milk (450–600 MPa) is more efficient in reducing immunoreactivity than thermal treatments unless they are conducted at high temperatures (85–99 °C). Loss of protein (amandin) solubility, mainly due to aggregation, rather than epitope destruction may account for the decreased immunogenicity [61]. The allergenicity of walnuts, however [62], decreases under the combined use of HP and heating (650 MPa, 100 °C, 15 min), thereby disclosing again the complicated issue of protein deactivation in allergenic samples through physical protocols.

Together with milk and egg-based allergens, the allergenicity of shrimp is a serious concern as well. While thermal treatment (roasting) reduces allergenicity by inducing protein unfolding, most epitopes are linear and bioassays in mice resulted in anaphylactic responses similar to those caused by raw shrimp protein. However, the combined action of roasting and pressure processing reduces significantly specific antibodies, mast cell degranulation, and other vascular effects with respect to mice fed with the raw protein [63]. Pressure sterilization is thought to cause protein aggregation, hiding digested epitopes inside the three-dimensional structure. The hypoallergenicity can also be ascribed to the low binding frequency of IgE to troponin C. 

Allergy to fish is a quite common phenomenon, albeit allergenicity to processed fish and seafood is often rare. This can be exemplified by tuna species as patients do not show allergic responses to canned tuna, even if occasional cases have been reported [64]. Fish parvalbumin is actually a panallergen that is responsible for cross-reactivity among a variety of fish species. The major allergen in tuna fish is parvalbumin (Thu a 1), which is highly stable toward thermal treatment. HP treatments of yellowfin tuna, for instance, in the range from 200 to 600 MPa for short times (5 min) have been reported as a method to extend the shelf-life of this product during storage (between 4 and 15 °C) [65,66]. Pressure not only reduces the bacterial load but also retards the loss of volatile nitrogen and histamine.

### 2.3. Pulsed Electric Fields

Pulsed electric fields (PEF) represent likewise an alternative to conventional thermal processing, especially when operating at high fields (>20 kV/cm) for short times (μs–ms). Applications range from extraction and recovery of nutritional substances to drying, freezing, and pasteurization processes, or detoxification, among others [67,68]. Compared with other physical methods, the assessment of PEF on allergenicity is still in its infancy and no clear-cut conclusions have been attained. Even if experimental setups involve essentially the coupling of a PEF generator to batch or continuous reactors, the outcomes are significantly influenced by parameters such as field intensity, voltage, or frequency. It seems that PEF at low electric strength (<10 kV/cm) does not induce any structural change of allergens, unlike the effects caused by HP or thermal processing [69]. However, the ability of ovalbumin to bind IgE and IgG1, together with the release rate of β-hexosaminidase is substantially inhibited at 10 kV/cm, while no effect takes place in the absence of electric pulses. These effects point to conformational changes and masking of sensitized epitopes that globally reduce allergenicity [70]. In addition, PEFs have been shown to inhibit the formation of Maillard products with respect to thermal treatment [71].

### 2.4. Ultrasound and Microwave Irradiations

The use of power ultrasound, with usual acoustic frequencies between 20 kHz and 500 kHz, is clearly an enabling technique that has gained considerable interest and application in food processing [72,73]. Most physical and chemical effects arise from cavitational collapse, i.e., the rapid formation, growth and implosion of microbubbles in liquids, thus generating enough kinetic energy, accompanied by shear forces and formation of reactive species that trigger subsequent reactions. Low-frequency sonication and short-time heating can be combined to ensure the stability of protein-based dairy formulations, for which prolonged heating causes thickening or gelling [74]. Ultrasound can promote the denaturation of proteins, albeit long irradiation times also lead to molecular degradation. Enzyme inactivation is dependent on acoustic parameters, especially frequency and power density, together with enzyme type, temperature, concentration, and pH. Deactivation of enzymes that degrade food quality can be accomplished by ultrasound alone or in conjunction with other processes such as mild heating and low pressure to shorten the radiation time [75]. 

As expected, cavitation can alter the supramolecular organization of biomolecules in the vicinity of microbubbles and induce aggregation through noncovalent interactions. This phenomenon appears to be responsible for changes in the properties of bean proteins [76]. Short-time sonication has been reported to modify the secondary structure of fruit proteins, namely loss of α-helices and increase in β-sheets, which cause a reduction in the IgE binding while enhancing protein digestibility [77].

Sonication (25-kHz probe, 900 W, 20 °C) of milk casein in the presence of a nonionic emulsifier (tween 80) at different concentrations results in colloidal casein with high transparency. Transmission electron microscopy (TEM) reveals how ultrasound disrupts the proteinaceous material and leads to particles with diameters less than 100 nm (Figure 5). ELISA assays show that the IgE-binding ability of such colloidal casein decreases, whereas the LAD_2_ mast cell line degranulation demonstrates the hypoallergenic character of ultrasound-treated casein as well. Such properties lasted for more than 30 days [78].

Unlike ultrasound, microwave (MW) irradiation constitutes an efficient thermal activation based on dielectric heating, involving the alignment of polar molecules in a rapidly oscillating electric field. This low-energy radiation can likewise modify the textural organization and molecular conformation of proteins. Reduction in the allergenicity of tropomyosin (up to 75%) present in shrimp has been described after MW heating at 125 °C for 15 min [79]. Although ultrasound (lacking quantum character) and MW (electromagnetic radiation) are very different in physical nature; synergistic effects have been observed by coupling both fields in simultaneous or sequential modes. Enhancements caused by the juxtaposition of such radiations derive from a fast thermal activation, while ultrasonic agitation improves mass transfer in heterogeneous mixtures [80]. Sequential MW and sonication have been applied to peach lipid transfer protein, which causes severe allergic reactions in sensitive patients. The protein in question is quite resistant to thermal processing and proteolysis. Unfortunately, the combined radiation is insufficient to reduce significantly IgE binding, as autoclave treatment does not decrease protein allergenicity either [81]. The effects of either MW or US on the secondary structure of egg white protein are chiefly influenced by temperature and irradiation times. Avidin activity decreases by MW heating from 60 to 80 °C (1 min vs 5 min-irradiation) and the unordered structures remain constant. Ultrasonic processing increases the proportion of β-sheets after 1 min-irradiation compared to the silent control, although the secondary structure content does not change appreciably after prolonged sonication [82].

## 3. Chemical/Biochemical Deactivation

### 3.1. Chemo-Enzymatic Treatments

Immunogenic responses can be attenuated by means of peptides, notably enzymes, often combined with other denaturation methods like heating or high pressure. Representative examples include mild thermal treatment and enzymatic hydrolysis with endoproteases on eggs. The modified product exhibits less allergenicity, in terms of decreased ovalbumin-specific IgE and IgG1 responses than untreated eggs in animal models [83]. Likewise, a hypoallergenic matrix can be obtained by pepsin-based hydrolysis of egg white proteins (ovalbumin, lysozyme and ovomucoid). The ovalbumin hydrolysate exhibits promising applicability toward peptide-based immunotherapy on the basis of cytokine production, although selective peptide fractionation does not increase the immune efficiency relative to the whole hydrolysate. This could be attributed to the presence of other peptides or adjuvants in the matrix [84]. Peptide-based approaches are also suitable for oral treatment because small peptides that contain T-cell epitope sequences can however avoid cross-reactions with IgE. Thus, regarding α_S1_-casein (Bos d 9), a prevalent allergen found in cow’s milk, T-cell epitopes have been assessed using overlapping peptides from the whole α_S1_-casein sequence. Accordingly, casein hydrolysates obtained by means of alcalase at alkaline pH result in a formulation with reduced antigenicity, yet retaining immunogenicity. The effect can be ascribed to less aggregation and precipitation through protein hydrolysis [85].

Ovomucoid is particularly unwilling to undergo enzyme hydrolysis and, as already mentioned, this protein constitutes a major allergenic compound of hen egg, which can inhibit the action of proteolytic trypsin as well. This undesired effect has been minimized through a covalent modification of ovomucoid (at the Gln 115 residue) with microbial transglutaminase in the presence of a dansylalated diamine (monodansyl cadaverine). Thermal incubation of ovomucoid is also instrumental to facilitate the specific binding to the enzyme [86]. A related two-step strategy employed for milk proteins involves the partial hydrolysis of whey proteins and casein with different proteases, followed by repolymerization with microbial transglutaminase, giving rise to significant structural modifications. Whereas whey partial hydrolysates with chymotrypsin, and thermolysin still show ca. 80% and 20% of the parent immunogenicity, repolymerization decreases such figures to 45 and 5%, respectively. Hydrolysis with trypsin reduces immunoreactivity as well (30%), but the repolymerized material remains essentially unaffected. In striking contrast, the enzymatic treatment of casein does not afford a hypoallergenic material compared to the untreated protein [87]. Cross-linking with more than one enzyme is envisaged as an additional approach in the case of allergens with high rates of cross-reactivity. Thus, structural modification of tropomyosin has been conducted with transglutaminase and tyrosinase in order to obtain a reticular matrix, which shows reduced allergenicity based on ELISA (enzyme-linked immunosorbent assay) and cell degranulation assays. A sensitized mouse model evidences the hypoallergenic character of enzyme-treated tropomyosin with reduced IgE response, yet maintaining the IgG-induction ability. Furthermore, transglutaminase appears to be more efficient than tyrosinase in regulating T-cell proliferation [88].

Mixed methods involving chemo-enzymatic treatments and physical deactivation represent further advances in OIT, albeit such actions may also result in lengthier protocols and extra costs. Structural modification of ovalbumin and egg white protein has been undertaken by combining transglutaminase as a cross-linking agent and high pressure (400 MPa at 40 °C for 30 min), although the resulting matrices do not show any significant hypoallergenicity relative to native proteins [89]. This result indicates that allergenic tests are compulsory because even substantial molecular alterations of specific proteins do not correlate with a lower immunogenicity. However, extensive chemical modification of α-lactalbumin through glycation, phosphorylation and acetylation of specific amino acid residues leads to reduced IgE and IgG binding capacity with release of histamine and IL-6. The net effect seems to be the masking of linear epitopes, which in turn derives from the modification of both active reaction sites and the native conformation [90]. 

A related structural alteration of milk protein has been conducted by means of enzymatic hydrolysis with immobilized trypsin and chymotrypsin, followed by thermal glycation with 10-kDa dextran, the latter exploiting a subsequent covalent modification via the Maillard reaction [91]. Likewise, an ultrasonic-ionic liquid pretreatment coupled with protease (either papain or alcalase) hydrolysis of whey proteins results in lower (82–88%) antigenic responses of the hydrolysates with respect to untreated milk proteins [92].

Not only purified enzymes but also microorganisms have been employed in recent years to achieve the proteolysis of food proteins. Four cows’ milk proteins, namely α-lactalbumin, β-lactoglobulin, α- and β-caseins, fermented by *Lactobacillus casei* 1134 exhibit reduced allergenicity based on IgE-binding inhibition, which decreases further in a simulation of gastrointestinal digestion [93,94]. Site-directed mutagenesis of allergenic ovomucoid (Gal d 1) by disruption of two cysteine residues at C192 and C210 positions in domain III of the protein results in diminished IgE reactivity, when the mutated and non-mutated proteins were expressed in *E. coli* and analyzed against a set of sera from egg-allergic patients [95].

### 3.2. Use of Phytochemicals

An inherently chemical treatment targeting the allergenic protein involves the use of naturally-occurring substances, which upon covalent/non-covalent interactions at the active site, will be able to misfold the native structure to some extent. Among chemical modifiers, polyphenols represent privileged scaffolds owing to their broad distribution in plants, often edible ones, and produced as secondary metabolites [96]. Recent research witnesses the positive effect of such substances on gut microbiota [97], the prevention of degenerative disorders [98], and the treatment of polycystic ovary syndrome [99]. Further health-promoting benefits are associated with the antioxidant and anti-inflammatory properties of polyphenols [100]. The former is also linked to the scavenging of Maillard-derived radicals and Strecker degradation products, including toxic compounds like acrylamide [39]. Also, a recent study provides new insights into the molecular mechanisms underpinned by polyphenols, by considering that the protein targets of such substances cluster in specific neighborhoods of the human interactome. Thus, experimental evidence now supports that rosmarinic acid inhibits platelet aggregation and α-granule secretion through inhibition of protein tyrosinase phosphorylation. This conceptual framework should be a starting point to integrate predictions on food metabolism, drug interactions or bioavailability [101].

Polyphenolic compounds have a high affinity to bind proteins, thereby generating soluble and insoluble complexes with altered functionality. The interaction between polyphenols and proteins present in food can take place by forming both covalent and non-covalent bonds. Interactions of flavonoids with milk proteins, for instance, could be an effective method to deliver flavonoids to the gut because, as mentioned above, numerous studies suggest that the flavonoid-microbiota interaction can also modulate cytokine-induced inflammation processes. In a representative study, the interaction of bovine β-lactoglobulin (BLG) with a few flavonoids, namely luteolin, myricetin, and hyperoside (Figure 1), the latter being the 3-*O*-galactoside of quercetin, has been reported en route to hypoallergenic bioconjugates [102]. In all cases, the best methods to monitor the structural changes on the secondary and tertiary structure of food proteins lie in FT-IR and UV/fluorescence spectroscopies, as they unravel significant changes in both intensity and shift of absorption bands. Thus, as shown in Figure 6, conjugation of BLG with the aforementioned flavonoids induces modification of amide I bands and others, whereas fluorescence spectroscopy points to conformational changes around specific amino acid residues. 

Although strong non-covalent interactions (i.e., H-bonding, ion-dipole) can account for such patterns, flavonoids are susceptible to oxidation, by forming quinone or semiquinone intermediates, which can further undergo covalent reaction with nucleophilic end-groups of protein side chains. Protein-flavonoid interactions are thought to lower α-helix content leading to more unfolded structures that correlate with decreased levels of IgG/IgE binding to BLG and hence lower allergenicity [102]. Likewise, the covalent interaction between chlorogenic acid and ovalbumin has been reported [103], as well as conjugation with whey proteins, which affect their allergenicity and digestibility. Whey protein isolates also exhibit enhanced emulsifying, foaming, and antioxidant properties after conjugation, although only in-vitro reduction of allergenicity has been evaluated [104].

A series of hypoallergenic peanut protein-polyphenol matrices with potentiality for OIT have been created [105]. The strategy employed represents an interesting and useful variation that would enable to revalue agrifood wastes in the industry, such as trimmings, peelings, stems, seeds, shells, bran, as well as residues remaining after extraction of oil, starch, sugar, and juice [106]. Citrus peel, and orange peel in particular, is noticeable because it accounts for about 50–60% of world citrus production and its elimination by conventional methods is insufficient [107]. Fruit peel is rich in a range of biomass-derived natural products such as cellulose, hemicellulose, and lignin, along with peptides and polyphenolics [108]. By upcycling vegetable wastes and converting them into added-value ingredients such as hypoallergenic matrices, food chemistry also adheres to the principles of circular and sustainable chemistry [109].

Food formulations containing proteins and polyphenols from berry extracts have been pre-aggregated into stable colloids prior to use. While phytochemicals remain highly bioavailable, the proteins do not undergo self-aggregation into clusters or react with other food ingredients. Interestingly, the reactive allergenic epitopes of some proteins are efficiently blunted by binding with polyphenols, thus altering their allergenicity. Likewise, both cytokine and chemokine production, characteristic of allergic reactions, are blocked as well [110]. In a related study, stable complexes from peanut proteins and either cranberry or blueberry pomace-based polyphenols exhibit in-vitro inhibition against IgE binding and mast cell degranulation. Quercetin, both as an isolated aglycon and glycoside derivative, is the main polyphenol bound to peanut proteins, which opens the door to a selective formulation containing this specific protein modifier [111]. 

As unveiled by structural studies conducted on protein-phenol interactions, like that performed on the caffeic acid-ovalbumin complex, changes in surface hydrophobicity, which globally modify secondary and tertiary structures, are the driving binding force. An increase in particle size upon complexation is also consistent with the role of caffeic acid as cross-linking agent [112]. Covalent conjugates between ovalbumin and epigallocatechin-3-gallate, have also been generated by radical and base-mediated reactions. Under radical conditions, conjugation with Cys74 and Glu347 residues occurs, whereas the alkaline reaction proceeds at Cys74 alone. The unfolded protein has increased digestibility, antioxidant, and foaming properties. As expected, IgE binding decreases as well through conjugation [113].

### 3.3. Allergoids

The search for a better benefit/risk balance of immunotherapy treatments has led to the emergence of polymerized extracts, specifically the so-called allergoids [114]. Allergoids are chemically modified allergens en route to hypoallergenic preparations that maintain the immunogenic response (i.e., stimulation of the immune system) while exhibiting less adverse reactions. The allergoids of polymerized protein-containing extracts are obtained by conjugation with glutaraldehyde or formaldehyde affording both intra- and intermolecular protein linking, thereby hiding the epitopic and catalytic areas which results in loss of IgE epitopes, yet maintaining T epitopes [115]. These features allow the administration of allergoids in high doses for short times [116]. Further modifications have also been introduced to improve the therapeutic ability such as polymerized allergens conjugated to nonoxidized mannan (PM-allergoids), which have been described as new products targeting dendritic cells to induce Tregs (regulatory T-cells) [117].

The advantages of these extracts are manifold and include improved efficacy, safety, the use of allergen mixtures, the possibility of vaccinating with a single extract against various antigenic sources, lower pharmaceutical costs, and reduction of school and work absenteeism, among others. The success of this treatment depends on the quality of the extract, which is based on standardization. The latter requires a series of well-established protocols, namely the identification of allergenic components, biological standardization, and quantification of major allergens. The allergen content in allergoids constitutes a major concern, although proteomics methods can help to spot them and quantify their relative and absolute amounts [118]. It is worth noting that allergoids have not yet been used as immunotherapy against food allergies, as avoidance and desensitization still represent the most widely employed treatments. However, common allergenic sources such as pollens and mites do indeed show cross-reactivity with food [119,120]. The effect of immunotherapy using pollen and house dust mite allergoids to treat associated food allergies is still little known [121,122], but it could be a promising way to treat food allergies in patients with allergen cross-reactivity. Overall, allergoids join the potential repertoire of chemical and physical processes targeting native proteins that are ultimately responsible for food allergy. Such modifications allow the development of standardized hypoallergenic extracts as immunotherapeutic agents for oral and non-oral delivery (Figure 7).

## 4. Conclusions

Food allergy is, if we may say so, the longest pandemic ever known by humans, rooted in our evolutionary history and continuous adaptation to edible substances and the environment. Being aware of the biomolecular pathways and immunological responses to allergens, we have now developed a useful toolbox to mitigate and/or circumvent the allergenicity caused by specific proteins in sensitized patients. Oral immunotherapy represents a valuable approach against food allergy, but it faces enormous challenges, especially prolonged treatments which should often be discontinued. Other allergen-attenuated delivery methods should be explored and developed while reducing psychological impacts as well. We have outlined a series of both physical and chemo-enzymatic protocols through recent and representative examples, all illustrating ways to deactivate the allergenicity and antigenicity of specific proteins. In any case, it is mandatory to check rigorously the in-vitro and in-vivo responses, as the as-generated structural modifications do not always involve the creation of a truly hypoallergenic material.

## Data Availability

Data are available from the authors on request.

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
