# Peer review of "Food Allergens: When Friends Become Foes—Caveats and Opportunities for Oral Immunotherapy Based on Deactivation Methods"

_nutrients, 2023, doi:10.3390/nu15163650_

Round 1
Reviewer 1 Report
The authors present a comprehensive look at protein modification methods in the context of food allergy and potentially oral immunotherapy.
Needs some organizations changes. For example line 57-72, why is cow's milk discussed here before talking about effects like bullying. Potentially paragraph should be moved to just before line 104 when OIT is discussed.
Minor Comments:
Line 92, discussion of T cells changes in OIT like reduction in CRTH2+ T cells.
Line 178 hypoallergenic vs allergenicity reduction?
High pressure Treatment - discussion of canned tuna?
Consider adding a few sentences in each section on protein modifications on how these changes can be/ have been implemented in OIT.
The language in the review is just wordy and often needs more clarity. Needs a thorough read through. Organization of sections could use improvement as well. The abstract is an example of how the language needs improvement. Tendency to jump from example to example with no clear transition. For example within a section (eg thermal treatment) the authors jump from cow's milk to fish with no clear transition or way to relate the examples.
Reviewer 2 Report
In this article, the findings of the recent studies on the appropriate modification of innocuous native proteins and the cancellation of potential risks in hypersensitive individuals, especially in children, were summarized. Also, this review described physical protocols conducted at lab and clinics.
The findings of previous reports were well summarized, and the illustrations seemed unique and concrete, and were therefore considered to assist in understanding the contents. This article was also well written, and seemed easy to read and understand, and might be nicely ordered. The contents of this review would probably be able to provide significant and useful information and knowledge. In addition, it would be quite interesting the description about the alterations in protein structures and allergenic capacities caused by physical, chemical and biochemical food processing. In particular, the description of OIT studies against egg-, milk- and peanut-caused allergies seemed quite interesting and instructive, and the information of the influence of the enzymatic treatment on allergenic materials might be possible to provide not a little information and knowledge for understanding and handling various food allergies. Therefore, it would be reasonable to consider that this manuscript might be worth publishing.
